# Dimensionality Reduction of SPD Data Based on Riemannian Manifold Tangent Spaces and Isometry

**DOI:** 10.3390/e23091117

**Published:** 2021-08-27

**Authors:** Wenxu Gao, Zhengming Ma, Weichao Gan, Shuyu Liu

**Affiliations:** 1School of Electronics and Information Technology, Sun Yat-Sen University, Guangzhou 510006, China; gaowx3@mail2.sysu.edu.cn (W.G.); ganwch5@mail2.sysu.edu.cn (W.G.); 2Public Experimental Teaching Center, Sun Yat-sen University, Guangzhou 510006, China; ljie@mail.sysu.edu.cn

**Keywords:** dimensionality reduction, tangent space, symmetric positive, definite matrices, isometry

## Abstract

Symmetric positive definite (SPD) data have become a hot topic in machine learning. Instead of a linear Euclidean space, SPD data generally lie on a nonlinear Riemannian manifold. To get over the problems caused by the high data dimensionality, dimensionality reduction (DR) is a key subject for SPD data, where bilinear transformation plays a vital role. Because linear operations are not supported in nonlinear spaces such as Riemannian manifolds, directly performing Euclidean DR methods on SPD matrices is inadequate and difficult in complex models and optimization. An SPD data DR method based on Riemannian manifold tangent spaces and global isometry (RMTSISOM-SPDDR) is proposed in this research. The main contributions are listed: (1) Any Riemannian manifold tangent space is a Hilbert space isomorphic to a Euclidean space. Particularly for SPD manifolds, tangent spaces consist of symmetric matrices, which can greatly preserve the form and attributes of original SPD data. For this reason, RMTSISOM-SPDDR transfers the bilinear transformation from manifolds to tangent spaces. (2) By *log* transformation, original SPD data are mapped to the tangent space at the identity matrix under the affine invariant Riemannian metric (AIRM). In this way, the geodesic distance between original data and the identity matrix is equal to the Euclidean distance between corresponding tangent vector and the origin. (3) The bilinear transformation is further determined by the isometric criterion guaranteeing the geodesic distance on high-dimensional SPD manifold as close as possible to the Euclidean distance in the tangent space of low-dimensional SPD manifold. Then, we use it for the DR of original SPD data. Experiments on five commonly used datasets show that RMTSISOM-SPDDR is superior to five advanced SPD data DR algorithms.

## 1. Introduction

The past few decades have witnessed the rapid development of machine vision, in which machine learning methods based on different mathematical platforms take various forms of image inputs. The most common form is the vector data flattened by a two-dimensional gray image or multi-channel color image [1,2,3]. This method is rough and simple on data processing and, although it has achieved a certain effect on some models, its problem is destroying the inherent position information and geometry structure of images. Another typical form is to keep the two-dimensional or higher-dimensional tensor data of images. There are lots of learning methods that take this data form as inputs [4,5,6] and the mathematical platform they all use involves tensor algebra. In addition, the convolutional neural network [7,8], as a branch of deep learning, has attained great achievements in processing tensor form for image classification. Though the position and statistics information of original image pixels are well maintained with tensor data as inputs, these methods cannot excavate the inherent geometric relations from the manifolds on which the data are located. On the other hand, taking the convolution neural network as an example, the excessively high computational complexity is also a drawback of the methods using tensor data. The last form is symmetric positive definite (SPD) data. Due to their ability to capture higher-order statistical information and excellent feature expression capacity, SPD data have been widely concerned and applied. Examples of mature applications include joint covariance descriptors for image sets and video frames [9,10,11], region covariance matrices for posture analysis [12,13], pedestrian detection [14,15] and texture discrimination [16,17,18]. Although SPD data provide many advantages and good properties for machine vision tasks, the applications are faced with certain challenges:SPD manifold is not a vector space and does not have the concepts of metric and inner product itself, so the traditional algorithms based on Euclidean space are not feasible here. It is not only unreasonable but also inadequate to apply the principles of Euclidean spaces to analyze SPD manifolds, for example, to directly use the Frobenius inner product to measure the similarity and dissimilarity between two SPD matrices.If the original dimensionality of SPD data is relatively high, there must be lots of redundant information, which not only affects the calculation speed, but may also affect the discrimination effect. In addition, it is well known that high-dimensional spaces can lead to curse of dimensionality, which is embodied in the exponential increase in the geometric volume of the space as the dimensionality increases, making available data extremely sparse. Further, sparsity is a problem for any method that requires statistical significance. Moreover, the high-dimensional feature space is of little significance to the distance metric. Since most classifiers rely on Euclidean distance or Mahalanobis distance, classification performance decreases with the increase in dimensionality.

In order to overcome challenge (1), we can equip the SPD manifold with a Riemannian metric to explain the geometric properties of SPD matrices by the Riemannian geometry. The similarity and dissimilarity between SPD samples can be effectively calculated by the derived corresponding geodesic distance under the assigned Riemannian metric. Some outstanding studies [19,20,21,22] have achieved satisfactory results in this regard. Another treatment method is to embed SPD manifolds to reproducing kernel Hilbert spaces (RKHS) by kernel tricks, then adopt the kernel-based learning methods [21,23,24]. For problem (2), the general solution is dimensionality reduction (DR), or low-dimensional embedding. 

This paper proposes a DR algorithm of SPD data based on Riemannian manifold tangent spaces and isometry (RMTSISOM-SPDDR). Different from existing supervised DR methods, our algorithm is unsupervised, which greatly saves the costs due to the fact that the labelling process for supervised learning methods is bound to consume time and effort. In summary, our contributions lie in the followings: (1)By using Riemannian manifold geometry, high-dimensional SPD data are mapped to the tangent space at the identity matrix, which makes linear operations possible and preserves the data form and attributes of original SPD data to the maximum extent.(2)We embedded the mapped data into a simple low-dimensional tangent space through bilinear transformation, so as to effectively avoid the problems caused by curse of dimensionality such as over-fitting and high calculation costs.(3)The bilinear transformation is determined by the proposed isometric criterion to maintain the distance relationship before and after DR. This is an unsupervised process, no manual labeling is required, which greatly saves labor and time costs.

The structure of this paper is as follows: In the Section 2, we introduce relevant preliminary knowledge needed for this paper, including notations, basic concepts of Riemannian manifolds. In the Section 3, some related works are briefly summarized. The Section 4 is our main part; in this section, we explain the proposal and solution of RMTSISOM-SPDDR in detail, with the algorithm flow and complexity analysis. The Section 5 mainly carries on comparative analysis of some advanced algorithms and introduces the connections and differences between them and RMTSISOM-SPDDR. In Section 6, sufficient experiments are conducted to verify the superiority of RMTSISOM-SPDDR. In Section 7, we make a summary of the whole paper.

## 2. Notations and Preliminaries

### 2.1. Notations

The basic notations in the paper are presented in Table 1.

### 2.2. Preliminaries

Riemannian manifolds are based on the concepts of differential manifolds and tangent spaces. For the content of differential manifolds and tangent spaces, please refer to the Appendix A. A generalized differential manifold is a topological space, but there is no distance metric or linear operations between its elements. Tangent spaces of a differential manifold support linear operations, but lack of distance metric between tangent vectors. In order to support machine learning, the differential manifold must be equipped with a Riemannian metric to make it a Riemannian manifold.

The Riemannian metric is a symmetric, positive definite, smooth second order tensor field of a differential manifold. Let $\Gamma_{2}\left(T_{p}\left(M\right)\right)$ represent the second order tensor space of the tangent space $T_{p}\left(M\right)$, where $\Gamma_{2}\left(M\right)=\underset{p\in M}{\cup }\Gamma_{2}\left(T_{p}\left(M\right)\right)$. Then the second order tensor field is defined as $\omega_{2}:M\to \Gamma_{2}\left(M\right)$, for any p∈M, $\omega_{2}\left(p\right)\in \Gamma_{2}\left(T_{p}\left(M\right)\right)$. If $\omega_{2}$ satisfies the following properties, for any $\Phi_{p},\Theta_{p}\in T_{p}\left(M\right)$

(1) symmetric: $\omega_{2}\left(p\right)\left(\Phi_{p},\Theta_{p}\right)=\omega_{2}\left(p\right)\left(\Theta_{p},\Phi_{p}\right)$

(2) positive definite:

$\omega_{2}\left(p\right)\left(\Phi_{p},\Phi_{p}\right)\geq 0$ and $\omega_{2}\left(p\right)\left(\Phi_{p},\Phi_{p}\right)=0$ ⇔ Φ=0

(3) smooth: $\omega_{2}$ is smooth on M

Then, $\omega_{2}$ is called a Riemannian metric on M and $\left(M,\omega_{2}\right)$ a Riemannian manifold.

By utilizing the Riemannian metric $\omega_{2}$, the length of any smooth curve ρ:(−ε,ε)→M on M can be defined as
(1)$$L\left(\rho \right)=\int_{-\varepsilon }^{\varepsilon }\sqrt{\omega_{2}\left(p\left(t\right)\right)\left(\rho^{\prime }\left(t\right),\rho^{\prime }\left(t\right)\right)}dt$$
Then, the geodesic distance between any two points on M is further defined, for any p,q∈M, as
(2)geoD(p,q)=min{L(ρ)|ρ:(−ε,ε)→M,ρ(−ε)=p,ρ(ε)=q}

Since the symmetric positive definite second order tensor field $\omega_{2}\left(p\right)$ on the tangent space $T_{p}\left(M\right)$ is the inner product of $T_{p}\left(M\right)$, once the Riemannian metric is defined, the inner product of the tangent space at each point of the Riemannian manifold is defined. Hence, tangent spaces of Riemannian manifolds are inner product spaces, which are finite-dimensional, namely, Hilbert spaces.

## 3. Related Works

In this section, we briefly review related works on discriminant analysis and dimensionality reduction (DR) of SPD manifolds. In recent years, the wide applications of multimedia and big data make SPD data occur in more and more occasions and attract attention of an extensive range. As mentioned in the Section 1, the applications of SPD data face some challenges, the most prominent of which are caused by the high dimensionality, for example, the overfitting caused by the case that the sample size is far smaller than the data dimensionality, the increase of computational complexity and the information redundancy caused by high-dimensional data. All these problems seriously affect the effective and reliable applications of SPD data on various occasions. In order to effectually solve above problems, the processing methods of SPD data in almost all relevant studies can be categorized into three kinds. First, the original SPD manifold is embedded into RKHS by an implicit mapping to utilize various kernel learning methods. Second, in order to use the machine learning models built on Euclidean spaces or to perform linear operations on SPD, the original SPD manifold is flattened to a Euclidean space, so as to make use of Euclidean methods. Third, by learning a mapping, the original SPD manifold is directly projected into a lower-dimensional one. These three methods can be said to be the basic methods. Other methods are the derivation or combination of these methods.

There is a lot of research on embedding SPD data into RKHS. This method can not only overcome the defect that original SPD manifolds do not support linear operations, but also be applicable to different occasions by choosing different kernel functions due to the diversity of kernel functions. Wang et al. [24] proposed a discriminant analysis method of multiple manifolds. It first constructs image sets into an SPD manifold and a Grassmann manifold, then embeds the obtained SPD manifold and Grassmann manifold into RKHS by Riemannian kernel and projection kernel, respectively, and, finally, carries out weighted discriminant analysis on them. A kernel-based sparse representation of Lie group and dictionary learning were proposed in [21] and successfully applied in face recognition and texture analysis. Huang et al. [25] mapped the source Euclidean space and Riemannian manifold to a common Euclidean subspace, which is implemented by a corresponding higher-dimensional RKHS. Through this mapping, the problem of learning the metric cross two heterogeneous spaces can be reduced into the problem of learning the Euclidean metric in the target Euclidean space. One of the advantages of using the kernel methods mentioned above is the multiplicity and selectivity of the kernel. However, choosing an impertinent kernel often results in performance degradation. In [26], a kernel learning method is used to overcome the kernel selection problem of classification on the Riemannian manifold. It presents two guidelines for jointly learning the kernel and classifier with the multi-kernel learning method.

Flattening SPD data into Euclidean forms is also a very classical method. Based on a spatio-temporal covariance descriptor, Sanin et al. [27] proposed a new motion and gesture recognition method, as well as a weighted Riemannian locally preserving projection method. The method requires linear operations; the first thing to do in this work is to perform tangential transformation on the spatio-temporal covariance descriptor, namely SPD data. Vemulapalli et al. [28] focused on logarithmic Euclidean Riemannian geometry and proposed a data-driven method for learning the Riemannian metric of SPD matrices. Huang et al. [9] used neural network structure. In this work, the input SPD matrix is converted into a more ideal SPD matrix by the design of the bilinear mapping layer and the nonlinear activation function is applied to the new SPD matrix by the eigenvalue correction layer. Finally, the data of the hidden are mapped to a Euclidian space in the output layer. A common problem with all of these works is that they inevitably destroy the geometry of SPD manifolds. In order to overcome this limitation, [10] proposed a method, LEML, to directly learn the logarithm of SPD matrix. By learning a tangent mapping, LEML can directly convert the matrix logarithm from the tangent space to another tangent space with more distinguishable information. Under the tangential mapping framework, the new metric learning can be reduced to the optimization seeking a Mahalanobis-like matrix, which can be solved by traditional metric learning techniques. Then, a more concise and efficient method, α-CML, was proposed based on the improvement of LEML in [29].

In many cases, SPD data are awfully high-dimensional, so how to effectively represent high-dimensional SPD data in low-dimensional space becomes particularly important. For SPD data, DR is usually performed by seeking a full column rank projection matrix. In [18], a direct DR method from high-dimensional SPD manifolds to low-dimensional ones is proposed. The projection generates a low-dimensional SPD manifold with maximum distinguishing ability through the orthogonal constraint and weighted affiliative similarity metric based on the manifold structure. Under the framework of the neural network, Dong et al. [30] made use of two layers (2D full connection layer and symmetric cleaning layer) to embed original SPD data into a subspace with lower dimension and better differentiation. On the basis of PCA [31], established a common component analysis (CCA) for multiple SPD matrices. This method attempts to find a low-dimensional common subspace but ignores the Riemannian geometry of the SPD manifold. From the perspective of geometric structure, Harandi et al. [32] used bilinear transformation to realize a supervised DR algorithm of SPD data and another unsupervised one. The supervised method gathers samples of the same class and separates samples of different classes at the same time to obtain a low-dimensional embedding representation with strong discrimination. In consideration of the maximum variance unfolding (MVU) algorithm [33], the unsupervised method maps SPD samples to a lower-dimensional SPD manifold in line with the criterion of maximum variance. Under the guidance of the nonparametric estimation theory, ref. [34] proposed a DR method of SPD which makes use of estimated probability distribution function (PDF) to construct an affinity matrix so as to achieve the maximum separability with respect to categories from the perspective of probability. 

## 4. The Proposed Method

In this paper, the proposed dimensionality reduction (DR) method based on tangent spaces and isometry first maps the original SPD data to the tangent space at the identity matrix; then, the bilinear transformation between manifolds is transferred to the tangent spaces. In the tangent space after DR, Euclidean metric is adopted to measure the distance between the data points after DR, while the bilinear transformation is determined by the isometric criterion. Specifically, we aim to keep the distance between data points in the tangent space after DR as close as possible to the corresponding geodesic distance on the original SPD manifold. With this idea in mind, we then derive the model.

### 4.1. The Isometric Transformation between SPD Manifolds and Their Tangent Spaces

SPD data are a kind of non-Euclidian data which can form a differential manifold under a certain structure. In particular, the tangent space of an SPD manifold is a symmetric matrix space, which can be regarded as the minimum linear extension set of the SPD manifold, that is, the minimum linear space containing the SPD matrices. In principle, machine learning algorithms developed in the symmetric matrix space are also suitable for its SPD subset.

As mentioned earlier, there is no distance metric on a differential manifold or its tangent spaces. A Riemannian metric must be endowed so as to obtain a Riemannian manifold. At present, there are several commonly used Riemannian metrics. The affine invariant Riemannian metric (AIRM) is adopted in this paper; for any $X\in Sym_{++}^{D}$, the inner product of the tangent space $T_{X}\left(Sym_{++}^{D}\right)$ at point X is defined as follows:(3)$${\langle \Phi ,\Theta \rangle }_{X}={\langle X^{-1/2}\Phi X^{-1/2},X^{-1/2}\Theta X^{-1/2}\rangle }_{F},\forall \Phi ,\Theta \in T_{X}\left(Sym_{++}^{D}\right)$$

Particularly, the identity matrix is an SPD and the affine invariant inner product of its tangent space $T_{I_{D}}\left(Sym_{++}^{D}\right)$ is as follows:(4)$${\langle \Phi ,\Theta \rangle }_{I_{D}}={\langle I_{D}^{-1/2}\Phi I_{D}^{-1/2},I_{D}^{-1/2}\Theta I_{D}^{-1/2}\rangle }_{F}={\langle \Phi ,\Theta \rangle }_{F},\forall \Phi ,\Theta \in T_{X}\left(Sym_{++}^{D}\right)$$

The Euclidean distance between tangent vectors derived from the inner product is
(5)$${\left\|\Phi -\Theta \right\|}_{I_{D}}^{2}={\langle \Phi -\Theta ,\Phi -\Theta \rangle }_{F},\forall \Phi ,\Theta \in T_{X}\left(Sym_{++}^{D}\right)$$

The geodesic distance on the SPD manifold defined by the AIRM is
(6)$$geoD\left(X,Y\right)={\left\|log\left(Y^{-1/2}{XY}^{-1/2}\right)\right\|}_{F},\forall X,Y\in Sym_{++}^{D}$$

Over here, $log:Sym_{++}^{D}\to Sym^{D}$. For any $X\in Sym_{++}^{D}$, by eigen-decomposition, we can have $X=Udiag\left(\lambda_{1},\ldots ,\lambda_{D}\right)U^{T}$. Hence,
(7)$$log\left(X\right)=Udiag\left(log\left(\lambda_{1}\right),\ldots ,log\left(\lambda_{D}\right)\right)U^{T}$$

It is obvious that log(X) has the same size as X. Besides, we note that the injection log(⋅) is monotonic, which does not change the order of eigenvalues. Hence, it realizes the transformation with the minimum change of properties compared with original data. 

If we intend to transfer learning tasks from the original SPD manifold to its tangent space, we first need to transform original data from the manifold to one of its tangent spaces. This paper selects the tangent space at identity $T_{I_{D}}\left(Sym_{++}^{D}\right)$ and log transformation. For any $X\in Sym_{++}^{D}$,
(8)$$geoD\left(X,I_{D}\right)={\left\|log\left(I_{D}^{-1/2}{XI}_{D}^{-1/2}\right)\right\|}_{F}={\left\|log\left(X\right)\right\|}_{F}$$
where ${\left\|log\left(X\right)\right\|}_{F}$ is the Euclidean distance between log(X) and the origin point. Hence, log is actually an isometric transformation; the geodesic distance between SPD data and the identity matrix is equal to the Euclidean distance between its tangent vector and the origin of the tangent space.

### 4.2. The Bilinear Transformation between Tangent Spaces

This paper studies the DR of SPD data. At present, the bilinear transformation of SPD matrices is the most commonly used method in DR algorithms of SPD data:$$f_{W}:Sym_{++}^{D}\to Sym_{++}^{d}$$

For $X\in Sym_{++}^{D}$,
(9)$$f_{W}\left(X\right)=W^{T}XW\in Sym_{++}^{d}$$
where $W\in R^{D\times d}$, $W^{T}W=I_{d}$, D≫d.

Obviously, $f_{W}$ is a transformation between two SPD manifolds. Since the SPD manifold is not a Euclidean space, it is difficult to directly solve $f_{W}$, no matter the modeling or optimization.

Due to the fact that the tangent space of an SPD manifold contains the SPD manifold itself, this paper proposes a DR model of bilinear transformation based on tangent spaces at identities of SPD manifolds:$$g_{W}:T_{I_{D}}\left(Sym_{++}^{D}\right)\to T_{I_{d}}\left(Sym_{++}^{d}\right)$$

For any $\Theta \in T_{I_{D}}\left(Sym_{++}^{D}\right)$,
(10)$$g_{W}\left(\Theta \right)=W^{T}\Theta W\in T_{I_{d}}\left(Sym_{++}^{d}\right)$$

Over here, $W\in R^{D\times d}$, $W^{T}W=I_{d}$, D≫d.

**Remark** **1:**

$g_{W}$

*is a transformation between tangent spaces of SPD manifolds, that is, a transformation between two Euclidean spaces. It is obviously much easier to build DR algorithms on Euclidean spaces.*


**Remark** **2:***If we just consider DR between tangent spaces, the constraint*$W^{T}W=I_{d}$*is unnecessary. Any matrix*$W\in R^{D\times d}$*can make*$g_{W}\left(\Theta \right)$*implement DR of the symmetric matrix*Θ. *However, our objective is to reduce data dimensionality for SPD matrices. By adding the constraint*$W^{T}W=I_{d}$*, it can be guaranteed that*${g_{W}|}_{Sym_{++}^{D}}=f_{W}$*on the SPD manifold.*

The advantage of the DR method based on tangent spaces of SPD manifolds proposed in this paper is that the DR between SPD manifolds is realized by the DR between Euclidean tangent spaces. A brief schematic diagram of the DR process is shown in Figure 1.

### 4.3. Dimensionality Reduction of SPD Data Based on Tangent Spaces of SPD Manifolds and Isometry

#### 4.3.1. Isometry Modeling

The DR scheme of SPD data based on tangent spaces of SPD manifolds and bilinear transformation described in Section 4.2 is only a framework. Moreover, it is necessary to determine the bilinear transformation matrix W according to a certain criterion. Different criteria generate different DR algorithms. In this paper, an isometric criterion is proposed to keep the distance between data points in the tangent space after DR close to the geodesic distance on the original SPD manifold as much as possible.

For a given set of data on an SPD manifold,
$$X_{i}\in {Sym}_{++}^{D},\;i=1,\ldots ,N$$

Since ${Sym}_{++}^{D}$ is a Riemannian manifold, the geodesic distance between two SPD matrices $X_{i}$ and $X_{j}$ (under AIRM) is
(11)$$g_{ij}=geoD\left(X_{i},X_{j}\right)={\left\|log\left(X_{j}^{-1/2}X_{i}X_{j}^{-1/2}\right)\right\|}_{F},\;1\leq i,j\leq N$$

After log transformation, the set of data is transformed to another set of data in $T_{I_{D}}\left(Sym_{++}^{D}\right)$:$$log\left(X_{i}\right)\in T_{I_{D}}\left(Sym_{++}^{D}\right),\;i=1,\ldots ,N$$

Afterwards, by the bilinear transformation between $T_{I_{D}}\left(Sym_{++}^{D}\right)$ and $T_{I_{d}}\left(Sym_{++}^{d}\right)$, we have
$$W^{T}log\left(X_{i}\right)W\in T_{I_{d}}\left(Sym_{++}^{d}\right),\;i=1,\ldots ,N$$

Because the tangent space $T_{I_{d}}\left(Sym_{++}^{d}\right)$ is a Euclidean space, the Euclidean distance between any two tangent vectors $W^{T}log\left(X_{i}\right)W$ and $W^{T}log\left(X_{j}\right)W$ is
(12)$$d_{ij}={\left\|W^{T}log\left(X_{i}\right)W-W^{T}log\left(X_{j}\right)W\right\|}_{F}={\left\|W^{T}\left(log\left(X_{i}\right)-log\left(X_{j}\right)\right)W\right\|}_{F}={\left\|W^{T}\Phi_{ij}W\right\|}_{F}$$
where $\Phi_{ij}=log\left(X_{i}\right)-log\left(X_{j}\right)$, 1≤i,j≤N.

The so-called isometric criterion is that the geodesic distance $g_{ij}$ is equal to the Euclidean distance $d_{ij}$.
(13)$$\sum_{i=1}^{N}\sum_{j=1}^{N}{\left(g_{ij}-d_{ij}\right)}^{2}\underset{W^{T}W=I_{d}}{\to }min$$

#### 4.3.2. Objective Function

This subsection derives the objective function of the model.

For convenience, we construct two distance matrices, as follows:(14)$$G=\left[\begin{array}{ccc} g_{11}^{} & \ldots & g_{1N}^{}\\ \vdots & \ddots & \vdots \\ g_{N1}^{} & \ldots & g_{NN}^{} \end{array}\right],D=\left[\begin{array}{ccc} d_{11}^{} & \ldots & d_{1N}^{}\\ \vdots & \ddots & \vdots \\ d_{N1}^{} & \ldots & d_{NN}^{} \end{array}\right]$$

According to the isometric criterion, W is selected to minimize the difference between the geodesic distance on the manifold and the distance in the tangent space at identity. Hence, the following model is obtained:(15)$$\hat{W}=\underset{W}{arg\;min}{\left\|G-D\right\|}_{F}^{2}\;\;\;\;s.t.\;W^{T}W{=I}_{d}$$

For $\forall g_{ij}\in G$ and $\forall d_{ij}\in D$, let $\varepsilon_{ij}={\left(g_{ij}-d_{ij}\right)}^{2}=g_{ij}^{2}+d_{ij}^{2}-2g_{ij}d_{ij}$; we expect that $g_{ij}$ and $d_{ij}$ are as close as possible and, replacing $g_{ij}$ with $d_{ij}$, we have $\varepsilon_{ij}\approx g_{ij}^{2}-d_{ij}^{2}$

Let
(16)$$G^{2}=\left[\begin{array}{ccc} g_{11}^{2} & \ldots & g_{1N}^{2}\\ \vdots & \ddots & \vdots \\ g_{N1}^{2} & \ldots & g_{NN}^{2} \end{array}\right],\;D^{2}=\left[\begin{array}{ccc} d_{11}^{2} & \ldots & d_{1N}^{2}\\ \vdots & \ddots & \vdots \\ d_{N1}^{2} & \ldots & d_{NN}^{2} \end{array}\right]$$

Then,
(17)$$\hat{W}=\underset{W}{arg\;min}{\left\|G-D\right\|}_{F}^{2}$$
can be represented in another form:(18)$$\hat{W}=\underset{W}{arg\;min}{\left\|G^{2}-D^{2}\right\|}_{F}^{2}$$

Let the centering matrix be Γ,
(19)Γ=(ID−1D(eeT))
where e=[1,1,…,1]T∈ℝD, ID is a *D* × *D* identity matrix.

Use (19) to center the objective function and obtain
(20)$$\begin{array}{ll} \hat{W} & =\underset{W}{arg\;min}{\left\|-\frac{1}{2}\Gamma \left(G^{2}-D^{2}\right)\Gamma \right\|}_{F}^{2}\\ & =\underset{W}{arg\;min}{\left\|-\frac{1}{2}\Gamma G^{2}\Gamma -\left(-\frac{1}{2}\Gamma D^{2}\Gamma \right)\right\|}_{F}^{2}\\ & =\underset{W}{arg\;min}{\left\|\rho_{G}-\rho_{D}\right\|}_{F}^{2} \end{array}$$

Over here, $\rho_{G}=-\frac{1}{2}\Gamma G^{2}\Gamma$, $\rho_{D}=-\frac{1}{2}\Gamma D^{2}\Gamma$.

It can be proved that $\rho_{G}$ and $\rho_{D}$ are both inner matrices [35], $d_{ij}^{2}={\left(\rho_{D}\right)}_{ii}+{\left(\rho_{D}\right)}_{jj}-2{\left(\rho_{D}\right)}_{ij}$.

According to the properties of linear algebra and matrix trace, Equation (20) can be further rewritten as
(21)$$\hat{W}=\underset{W}{arg\;min}\left[-2tr\left(\rho_{G}\rho_{D}{}^{T}\right)+{\left\|\rho_{G}\right\|}_{F}^{2}+{\left\|\rho_{D}\right\|}_{F}^{2}\right]$$

In consideration of
(22)$$d_{ij}={\left\|\Theta_{i}-\Theta_{j}\right\|}_{F}={\left\|W^{T}\left(log\left(X_{i}\right)-log\left(X_{j}\right)\right)W\right\|}_{F}\leq {\left\|X_{i}-X_{j}\right\|}_{F}$$
hence,
(23)$${\left\|\rho_{D}\right\|}_{F}^{2}\leq {\left\|\rho_{S}\right\|}_{F}^{2}$$
where
(24)$$S^{2}=\left[\begin{array}{ccc} s_{11}^{2} & \ldots & s_{1N}^{2}\\ \vdots & \ddots & \vdots \\ s_{N1}^{2} & \ldots & s_{NN}^{2} \end{array}\right],s_{ij}={\left\|X_{i}-X_{j}\right\|}_{F},\rho_{S}=-\frac{1}{2}\Gamma S^{2}\Gamma$$

It can be seen that the objective function has an upper boundary:(25)$$\begin{array}{ll} \hat{W} & =\underset{W}{arg\;min}\left[-2tr\left(\rho_{G}\rho_{D}{}^{T}\right)+{\left\|\rho_{G}\right\|}_{F}^{2}+{\left\|\rho_{G}\right\|}_{F}^{2}\right]\\ & \leq \underset{W}{arg\;min}\left[-2tr\left(\rho_{G}\rho_{D}{}^{T}\right)+{\left\|\rho_{G}\right\|}_{F}^{2}+{\left\|\rho_{S}\right\|}_{F}^{2}\right] \end{array}$$

Since $G^{2}$ and $S^{2}$ are unrelated to W, we can further simplify the objective function as
(26)$$\begin{array}{ll} \hat{W} & ={\underset{W}{arg\;min}}_{}^{}-2tr\left(\rho_{G}\rho_{D}{}^{T}\right)\\ & ={\underset{W}{arg\;min}}_{}^{}-2\sum_{i=1}^{N}\sum_{j=1}^{N}{\left(\rho_{D}\right)}_{ij}{\left(\rho_{G}\right)}_{ij} \end{array}$$

In consideration of
(27)$$2{\left(\rho_{D}\right)}_{ij}={\left\|\Theta_{i}\right\|}_{F}^{2}+{\left\|\Theta_{j}\right\|}_{F}^{2}-{\left\|\Theta_{i}-\Theta_{j}\right\|}_{F}^{2}$$
substitute (27) into (26),
(28)$$\begin{array}{ll} \hat{W} & ={\underset{W}{arg\;min}}_{}^{}-2\sum_{i=1}^{N}\sum_{j=1}^{N}{\left\|\Theta_{i}\right\|}_{F}^{2}{\left(\rho_{G}\right)}_{ij}+\sum_{i=1}^{N}\sum_{j=1}^{N}{\left\|\Theta_{i}-\Theta_{j}\right\|}_{F}^{2}{\left(\rho_{G}\right)}_{ij}\\ & ={\underset{W}{arg\;min}}_{}^{}-2\sum_{i=1}^{N}{\left\|\Theta_{i}\right\|}_{F}^{2}\sum_{j=1}^{N}{\left(\rho_{G}\right)}_{ij}+\sum_{i=1}^{N}\sum_{j=1}^{N}{\left\|\Theta_{i}-\Theta_{j}\right\|}_{F}^{2}{\left(\rho_{G}\right)}_{ij} \end{array}$$

On account of
(29)$$\sum_{j=1}^{N}{\left(\rho_{G}\right)}_{ij}=0$$
we have
(30)$$\begin{array}{ll} \hat{W} & ={\underset{W}{arg\;min}}_{}^{}\sum_{i=1}^{N}\sum_{j=1}^{N}{\left\|\Theta_{i}-\Theta_{j}\right\|}_{F}^{2}{\left(\rho_{G}\right)}_{ij}\\ & =argmin\sum_{i=1}^{N}\sum_{j=1}^{N}{\left\|W^{T}log\left(X_{i}\right)W-W^{T}log\left(X_{j}\right)W\right\|}_{F}^{2}{\left(\rho_{G}\right)}_{ij}\\ & =argmin\sum_{i=1}^{N}\sum_{j=1}^{N}tr\left(W^{T}\left(log\left(X_{i}\right)-log\left(X_{j}\right)\right)WW^{T}{\left(log\left(X_{i}\right)-log\left(X_{j}\right)\right)}^{T}W\right){\left(\rho_{G}\right)}_{ij} \end{array}$$

So far, we have derived the objective function of the model and what we need to do next is optimizing the function.

### 4.4. Solution to RMTSISOM-SPDDR

In this subsection, we solve the derived objective function. Obviously, this is a convex optimization problem, but this objective function does not correspond to a closed-form solution, so we are supposed to find its local optimal solution $\hat{W}$ through an iterative method.

Let the t^th^ iteration yield ${\hat{W}}_{t}$, then the result of the t+1^th^ iteration can be expressed as
(31)$$\begin{array}{ll} {\hat{W}}_{t+1} & =argmin\sum_{i=1}^{N}\sum_{j=1}^{N}tr\left(W^{T}\left(log\left(X_{i}\right)-log\left(X_{j}\right)\right){\hat{W}}_{t}{\hat{W}}_{t}{}_{t}^{T}{\left(log\left(X_{i}\right)-log\left(X_{j}\right)\right)}^{T}W\right){\left(\rho_{G}\right)}_{ij}\\ & =argmaxtr\left(W^{T}LW\right) \end{array}s.t.\;W^{T}W{=I}_{d}$$

Over here,
(32)$$L=-\sum_{i=1}^{N}\sum_{j=1}^{N}{\left(\rho_{G}\right)}_{ij}\left(log\left(X_{i}\right)-log\left(X_{j}\right)\right){\hat{W}}_{t}{\hat{W}}_{t}^{T}{\left(log\left(X_{i}\right)-log\left(X_{j}\right)\right)}^{T}$$

The eigenvalue decomposition of L is needed and W^t+1 is obtained by taking the eigenvectors corresponding to the maximum ***d*** eigenvalues. The entire algorithm flow of RMTSISOM-SPDDR is shown in Algorithm 1.
**Algorithm 1.** Procedures of RMTSISOM-SPDDR.**Input:*****N*** training samples X={X1,X2,…,XN}, Xi∈Sym++D, target dimension ***d*** and the maximum iteration times ***maxIter***.**Output:** bilinear transformation matrix W∈ℝD×d. 1: Map SPD data into the corresponding tangent space at identity using Eqution (7); 2: Construct Riemannian metric matrix using Equation (16); 3: Construct the inner product matrix ρG using Equation (20); 4: Initialize W^0=ID(:,1:d), namely, the first ***d*** columns of ID∈ℝD×D; 5: while t = 1: ***maxIter***       Calculate L using Equation (32);            Do eigen-decomposition of L to obtain the bilinear transformation matrix W^t;      end while 6: return W^t.

### 4.5. Complexity Analysis

In this subsection, we analyze the computational complexity of RMTSISOM-SPDDR. The complexity of the algorithm comes from five aspects—the mapping from the SPD manifold to its tangent space at identity, the construction of the Riemannian metric matrix, the calculation of the inner matrix $\rho_{G}$, the construction of L and the eigen-decomposition of L. First of all, the mapping from the SPD manifold to the tangent space at identity requires eigen-decomposition of each data point and taking the logarithm of its eigenvalues. This process can be realized by the built-in function eig in MATLAB and the time complexity of the implementation is $O\left(ND^{3}log\left(D\right)\right)$. Secondly, the construction of the Riemannian metric matrix requires calculating the geodesic distances of data pairs. Note, it is a symmetric matrix which needs to calculate N(N+1)/2 times. Hence, the corresponding time complexity is $O\left(N\left(N+1\right)/2\right)=O\left(\frac{1}{2}N^{2}+\frac{1}{2}N\right)=O\left(N^{2}\right)$. The calculation of $\rho_{G}$ involves, first, squaring each element of the Riemannian metric matrix, then multiplying it left and right by the centering matrix, so the time complexity of this procedure is $O\left(N^{2}+N^{4}\right)=O\left(N^{4}\right)$. Besides, the construction of L requires four matrix multiplications and $N^{2}$ matrix additions, the time complexity is $O\left(N^{2}\left(d^{2}D^{4}\right)\right)$. Finally, the complexity of the eigen-decomposition for L is $O\left(D^{3}log\left(D\right)\right)$. 

Since these five processes are cascaded, the final time complexity of RMTSISOM-SPDDR is $O\left(ND^{3}log\left(D\right)+N^{2}+N^{4}+N^{2}\left(d^{2}D^{4}\right)+D^{3}log\left(D\right)\right)$. 

## 5. Comparison with Other State-of-the-Art Algorithms

In this section, we introduce five state-of-the-art DR algorithms of SPD data, including PDDRSPD [34], PSSSD [36], DARG-Graph [37], SSPDDR [32] and LG-SPDDR [38], in summary. Here, we also point out the differences and connections between RMTSISOM-SPDDR and these algorithms. 

These DR algorithms for comparison all belong to supervised learning and affinity, in which data labels are needed. In contrast, RMTSISOM-SPDDR is unsupervised and on the basis of global isometry. Except for LG-SPDDR, the DR of all other comparison algorithms is based on direct bilinear transformation between SPD matrices, while the DR of RMTSISOM-SPDDR is based on the bilinear transformation between tangent vectors of SPD matrices. LG-SPDDR utilizes exp transformation for the dimensional-reduced tangent vectors to map them back to a low-dimensional manifold, which means its framework is different from ours.

### 5.1. SSPDDR

In SSPDDR [32], a supervised DR method of SPD data is proposed, where the metric $\delta \left(X_{i},X_{j}\right)$ is given to measure the similarity among SPD data. Moreover, the bilinear transformation is employed to the DR process: $$X\in Sym_{++}^{D}\Rightarrow W^{T}XW\in Sym_{++}^{d}$$

Over here, $W\in R^{D\times d}$, $W^{T}W=I_{d}$ and D≫d. We can easily know that, if X is an SPD matrix, $W^{T}XW$ is an SPD matrix as well.

In SSPDDR, the transformation matrix W can be determined as follows:(33)$$\sum_{j=1}^{N}\sum_{i=1}^{N}a\left(X_{i},X_{j}\right)\delta \left(W^{T}X_{i}W,W^{T}X_{j}W\right)\underset{W^{T}W=I_{d}}{\to }min$$

Over here, $a\left(X_{i},X_{j}\right)=a_{w}\left(X_{i},X_{j}\right)-a_{b}\left(X_{i},X_{j}\right)$ and
(34)$$a_{w}\left(X_{i},X_{j}\right)=\{\begin{array}{cc} 1 & X_{i}{\;\mathrm{and}\;X}_{j}\;are\;adjacent\;and\;have\;the\;same\;label\\ 0 & others \end{array}$$
(35)$$a_{b}\left(X_{i},X_{j}\right)=\{\begin{array}{cc} 1 & X_{i}{\;\mathrm{and}\;X}_{j}\;are\;adjacentbut\;have\;\mathrm{different}\;labels\\ 0 & others \end{array}$$

SSPDDR indicates that if $X_{i}$ and $X_{j}$ are adjacent and intraclass, their compactness is maximized in the process of DR, or else their separability is minimized. If $X_{i}$ and $X_{j}$ are not adjacent, their relationship will be neglected in DR.

### 5.2. PDDRSPD

In PDDRSPD [34], labeled SPD data $\left\{\left(X_{i},c_{i}\right)|i=1,\ldots ,N\right\}$ are given, in which $c_{i}$ is the label of $X_{i}$,.$c_{i}\in \left\{1,\ldots ,L\right\}$. 

For each class, PDDRSPD firstly calculates its mean vector and covariance matrix. Subsequently, based on nonparametric estimation, it estimates the Gaussian density functions $\left\{f_{c}\left(X\right)|c=1,\ldots ,L\right\}$.

The framework of PDDRSPD is similar to [32], but the affinity coefficients is quite different:(36)$$\sum_{j=1}^{N}\sum_{i=1}^{N}a\left(X_{i},X_{j}\right)\delta \left(W^{T}X_{i}W,W^{T}X_{j}W\right)\underset{W^{T}W=I_{d}}{\to }min$$

Over here $a\left(X_{i},X_{j}\right)=a_{w}\left(X_{i},X_{j}\right)-a_{b}\left(X_{i},X_{j}\right)$ and
(37)$$a_{w}\left(X_{i},X_{j}\right)=\{\begin{array}{cc} \frac{\left(\lambda_{w}-1\right)\left|f_{c}\left(X_{i}\right)-f_{c}\left(X_{j}\right)\right|}{\sum_{X\in N_{w}\left(X_{i}\right)}\left|f_{c}\left(X_{i}\right)-f_{c}\left(X\right)\right|} & X_{j}\in N_{w}\left(X_{i}\right)\\ 1 & i=j\\ 0 & others \end{array}$$
where c represents the common label of $X_{i}$ and $X_{j}$, $N_{w}\left(X_{i}\right)$ represents the set of $\lambda_{w}$ samples belonging to the same class as $X_{i}$ and farthest from $X_{i}$.
(38)$$a_{b}\left(X_{i},X_{j}\right)=\{\begin{array}{cc} \frac{\lambda_{b}f_{c_{j}}\left(X_{j}\right)}{\lambda_{s}\sum_{X\in N_{b}\left(X_{j}\right)}f_{c_{j}}\left(X\right)} & c_{j}\in C_{b}\left(X_{i}\right)\\ 0 & others \end{array}$$

Over here, $C_{b}\left(X_{i}\right)=\left\{c_{i_{1}},\ldots ,c_{i_{\lambda_{b}}}\right\}$ and
(39)$$\underset{\lambda_{b}}{\underset{︸}{f_{c_{i_{1}}}\left(X_{i}\right)\geq \ldots \geq f_{c_{i_{\lambda_{b}}}}\left(X_{i}\right)}}\geq \ldots \geq f_{c_{i_{L}}}\left(X_{i}\right)$$

$N_{b}\left(X_{j}\right)$ is the set of $\lambda_{s}$ SPD matrices belonging to the same class as $X_{j}$ and corresponding to largest Gaussian density values.

### 5.3. LG-SPDDR

In LG-SPDDR [38], a DR method for SPD data is proposed as
$$X\in Sym_{++}^{D}\Rightarrow Y_{W}=exp\left(W^{T}log\left(X\right)W\right)\in Sym_{++}^{d}$$
where $W\in R^{D\times d}$, D≫d. 

To be specific, LG-SPDDR is implemented by three procedures:
The given SPD data X are firstly mapped from the original SPD manifold $Sym_{++}^{D}$ to the corresponding tangent space $T_{I_{D}}\left(Sym_{++}^{D}\right)=Sym^{D}$ and turn into $log\left(X\right)\in Sym^{D}$;Subsequently, log(X) is dimensional-reduced by the bilinear transformation, transformed into another tangent space $T_{I_{d}}\left(Sym_{++}^{d}\right)=Sym^{d}$ and turns into $W^{T}log\left(X\right)W\in Sym^{d}$;Finally, through the *exp*-transformation, $W^{T}log\left(X\right)W$ is remapped back into a new low-dimensional SPD manifold $Sym_{++}^{d}$ and turns into $exp\left(W^{T}log\left(X\right)W\right)\in Sym_{++}^{d}$.

LG-SPDDR determines the transformation matrix W as follows:(40)$$\sum_{j=1}^{N}\sum_{i=1}^{N}a\left(X_{i},X_{j}\right)\delta \left(Y_{i}^{W},Y_{j}^{W}\right)\underset{W\in R^{D\times d}}{\to }min$$

Over here, the definition of affinity coefficients $a\left(X_{i},X_{j}\right)$ is similar to [32].

**Remark** **3:**
*In spite of the claim that LG-SPDDR is based on Lie groups and Lie algebra, it seems to be irrelevant to the multiplication of Lie groups and the Lie bracket of Lie algebra.*


### 5.4. PSSSD

PSSSD [36] is a supervised metric learning method. Both the bilinear transformation and label information are taken into account in the learning process. Its core idea is a framework for learnable metric, in which:

(1) A set of divergences measuring the similarity between two SPD matrices have been given:(41)$$D^{\left(\alpha_{m},\beta_{m}\right)}\left(X_{i}\|X_{j}\right)=\frac{1}{\alpha \beta }log\left(\left|\frac{\alpha_{m}{\left(X_{i}X_{j}^{-1}\right)}^{\beta_{m}}+\beta_{m}{\left(X_{i}X_{j}^{-1}\right)}^{\alpha_{m}}}{\alpha_{m}+\beta_{m}}\right|\right)$$
$$\alpha_{m}\beta_{m}\neq 0,\;\alpha_{m}+\beta_{m}\neq 0$$

Over here, $\alpha_{m}$ and $\beta_{m}$ are all learnable, m=1,…,M.

(2) It exploits multiple transformation matrices to map original SPD manifold to multiple submanifolds:$$W_{m}\in R^{D\times d},\;W_{m}^{T}W_{m}=I_{d},\;m=1,\ldots ,M$$

(3) To measure the difference between two SDP matrices $X_{i}$ and $X_{j}$, the learnable metric is defined as
(42)$$D^{\Theta }\left(X_{i}\|X_{j}\right)=d_{ij}^{T}\Phi d_{ij},\;d_{ij}=\left[\begin{array}{c} D^{\left(\alpha_{1},\beta_{1}\right)}\left(W_{1}^{T}X_{i}W_{1}\|W_{1}^{T}X_{j}W_{1}\right)\\ \vdots \\ D^{\left(\alpha_{M},\beta_{M}\right)}\left(W_{M}^{T}X_{i}W_{M}\|W_{M}^{T}X_{j}W_{M}\right) \end{array}\right]\in R^{M}$$

Over here, $\Phi \in R^{M\times M}$ is a learnable SPD matrix. Hence, entire learnable parameters can be represented by
(43)$$\Theta =\left\{\Phi ,W_{1},\ldots ,W_{M},\alpha_{1},\ldots ,\alpha_{M},\beta_{1},\ldots ,\beta_{M}\right\}$$

The metric $D^{\Theta }\left(•\|•\right)$ is finally specified as
(44)$$\frac{1}{\left|S\right|}\sum_{\left(i,j\right)\in S}y_{ij}max{\left(D^{\Theta }\left(X_{i},X_{j}\right)-\zeta_{S},0\right)}^{2}+\frac{1}{\left|ℋ\right|}\sum_{\left(i,j\right)\in ℋ}\left(1-y_{ij}\right)max{\left(\zeta_{ℋ}-D^{\Theta }\left(X_{i},X_{j}\right),0\right)}^{2}\underset{\Theta }{\to }min$$
where S represents the set of data pairs having the same label and ℋ represents the set of data pairs having different labels; $y_{ij}=\{\begin{array}{cc} 1 & {If\;X}_{i}{\;and\;X}_{j}\;are\;similar\\ 0 & otherwise \end{array}$, $\zeta_{S}$ and $\zeta_{ℋ}$ are similarity thresholds.

The objective of LG-SPDDR indicates that, if $X_{i}$ and $X_{j}$ are similar to each other and belong to the same class, the metric $D^{\Theta }\left(X_{i},X_{j}\right)$ is minimized. In contrast, if $X_{i}$ and $X_{j}$ are similar but belong to the different classes, $D^{\Theta }\left(X_{i},X_{j}\right)$ is maximized.

Strictly speaking, PSSSD is essentially not a DR algorithm. To compare it with RMTSISOM-SPDDR, the matrix $W=\left[\begin{array}{ccc} W_{1} & \ldots & W_{M} \end{array}\right]\in R^{D\times Md}$ is used for DR. 

### 5.5. DARG-Graph

Let g=N(μ,Σ) be a Gaussian distribution, $\mu \in R^{D}$ be its mean vector and $\Sigma \in R^{D\times D}$ be its covariance matrix. In DARG-Graph, a number of metrics $\delta \left(g_{i},g_{j}\right)$ to measure the difference between two Gaussian distributions $g_{i}$ and $g_{i}$ are given. 

In DARG-Graph [37], the DR of a Gaussian distribution is proposed as follows: $$g=N\left(\mu ,\Sigma \right)\Rightarrow g^{W}=N\left(W^{T}\mu ,W^{T}\Sigma W\right)$$
where $W\in R^{D\times d}$ and $W^{T}W=I_{d}$, D≫d. We can easily know that $g^{W}$ is a Gaussian distribution as well. 

The transformation matrix W is specified as follows:(45)$$\sum_{j=1}^{N}\sum_{i=1}^{N}a\left(g_{i},g_{j}\right)\delta \left(g_{i}^{W},g_{j}^{W}\right)\underset{W^{T}W=I_{d}}{\to }min$$

Over here, $a\left(X_{i},X_{j}\right)=a_{w}\left(X_{i},X_{j}\right)-a_{b}\left(X_{i},X_{j}\right)$ and
(46)$$a_{w}\left(g_{i},g_{j}\right)=\{\begin{array}{cc} e^{-\frac{\delta \left(g_{i},g_{j}\right)}{\sigma^{2}}} & \delta \left(g_{i},g_{j}\right)<\varepsilon \;\;and\;have\;the\;same\;label\\ 0 & others \end{array}$$
(47)$$a_{b}\left(g_{i},g_{j}\right)=\{\begin{array}{cc} e^{-\frac{\delta \left(g_{i},g_{j}\right)}{\sigma^{2}}} & \delta \left(g_{i},g_{j}\right)<\varepsilon \;\;but\;have\;different\;labels\\ 0 & others \end{array}$$

Note that RMTDISOM-SPDDR is a DR algorithm for SPD data, not for Gaussian distributions. When comparing our algorithm with DARG-Graph, we let vector μ=0. Σ in N(μ,Σ) is an SPD matrix. Hence, the DR of Gaussian distributions is converted to the DR of SPD data. 

## 6. Experiments

In this section, we compare RMTSISOM-SPDDR with five state-of-the-art algorithms by evaluating the performance of algorithms on five commonly used benchmark datasets. This section will introduce the datasets and the setting of experimental parameters, present the experimental results and further discuss the effectiveness of our algorithm.

### 6.1. Datasets Description and Experiment Settings

**Dataset overview:** Since we conduct experiments on five datasets and the processing of different datasets is not the same, it is not appropriate to list all descriptions here. We explain the specific description of each dataset one by one in the following corresponding subsections.

**Parameter setting:** The parameters that need to be adjusted for RMTSISOM-SPDDR include the target dimension d of the SPD data after DR and the maximum iteration number ***maxIter*** of iterative optimization. For d, the specific setting is related to the dimensionality of original SPD manifolds on which different datasets are located. It can be observed from the different experimental charts how we set it. For ***maxIter***, we uniformly set it as 50, in this paper, for three reasons. Firstly, in our experimental results, under different target dimensions on all datasets, RMTSISOM-SPDDR converged before 50 iterations consistently. Hence, too many iterations are uneconomical for the calculation cost. Secondly, it is insignificant to improve performance by more iterations, which may result in overfitting, making worse performance. Thirdly, although the optimization process iterated for 50 times, the optimal result may occur before the 50th iteration. Our program recorded the classification result of each iteration and we selected the optimal result as the final experimental result. In addition, the classifier used in the experiments was the kNN classifier. For other algorithms, parameter settings are all shown in Table 2.

### 6.2. The Experiment on the Motion Dataset

Our first experiment was conducted on the Motion dataset. In this subsection, we successively introduce Motion and the experimental development and present results of RMTSISOM-SPDDR and other algorithms on this dataset. Subsections for other datasets keep the same narrative structure and we do not repeat this description.

#### 6.2.1. The Description of Motion

Motion is derived from the human motion sequences in the HDM05 database [39], which was produced by a system based on optical mark technology. Experimenters worn a suit with 40–50 retro-reflective markers, which were tracked by 6–12 calibrated high-resolution cameras, resulting in clear and detailed motion capture data. The database consists of 14 kinds of motions, such as walking, running, jumping, kicking, etc. Partial samples of Motion are shown in Figure 2. Motion includes 306 image sets, each of which consists of 120 images. By the method in [24], they were made into 93 × 93 SPD matrices. We adopted the same partition method as [32] and obtained SPD matrices for training and testing, respectively, with the ratio of 2:3.

#### 6.2.2. The Experimental Results on Motion

On Motion, we carried out classification experiments for the proposed algorithm (RMTSISOM-SPDDR) and other five comparison algorithms, respectively, and tested the convergence performance of RMTSISOM-SPDDR. Subsequent experiments on other datasets also followed the same development and we do not repeat this description in the other subsections. 

From the classification results on Motion shown in Figure 3, we can see, under each target dimension selected, the classification accuracy of RMTSISOM-SPDDR is generally more than 70% and far ahead of all comparison algorithms. Especially under the target dimensions of 48, 43, 38 or 33, the accuracy of RMTSISOM-SPDDR is 6 percentage points higher than that of the comparison algorithm ranked second (LG-SPDDR). On the whole, the fluctuation of the classification accuracies of RMTSISOM-SPDDR under different target dimensions is relatively small compared with other algorithms.

To evaluate the convergence performance of RMTSISOM-SPDDR under different target dimensions on Motion, we obtained Figure 4. The loss function decreased the most in the second iteration, then became stable. Among these four curves, the convergence was the fastest when d=43 or d=53 and it basically converged at the fifth iteration. When d=33 or d=63, the loss function continued to decline in 50 iterations, but the rangeability was very small after 25 iterations. It was unnecessary to continue iterations after 50 iterations. On the one hand, the decline of the loss function in continuous iterations was insignificant, compared with the calculation cost paid. On the other hand, excessive pursuit of the decline of the loss function may lead to overfitting. When the embedding dimension was equal to 33, the loss value fluctuated greatly, but the overall trend still declined. RMTSISOM-SPDDR still shows satisfactory convergence performance on Motion. It is worth noting that the lower the target dimension, the smaller the absolute value of the loss function, which means that we need less computation.

### 6.3. The Experiment on the Traffic Dataset

#### 6.3.1. The Description of Traffic

The Traffic video dataset is derived from the UCSD traffic dataset [40], which contains 254 freeway video sequences recorded by a fixed camera from a freeway in Seattle. These sequences can be roughly divided into three traffic conditions (heavy, medium, light; see Figure 5). The three traffic conditions correspond to 44, 45 and 165 sequences, respectively. Each frame was converted to a grayscale image whose size was adjusted to 125 × 125 pixels. We used HoG features [41] to represent frames of a traffic video. In this way, one feature vector corresponded to one frame. In order to acquire the SPD data needed, the feature vectors were normalized and corresponding covariances were calculated. In the experiments on Traffic, we selected 190 sequences for training and the remaining 64 sequences as the test set.

#### 6.3.2. The Experimental Results on Traffic 

Figure 6 presents the results of the classification experiments on Traffic. The classification accuracies of RMTSISOM-SPDDR are not all the way ahead of other comparison algorithms. This is mainly reflected in the three dimensions of 45, 40 and 30, in which the accuracies of DARG are higher than ours. The possible reason might be that the scene is relatively complex, while DARG firstly adopts a Gaussian mixed model to extract data features, so as to obtain more discriminant information in high dimensions. However, in cases of low dimensions, the accuracies of DARG drop sharply, while ours still remain at a high level. For SPD data, we have already discussed the problems caused by their high dimensionality, so the DR of SPD data is the objective we are committed to. In such a case, RMTSISOM-SPDDR shows superiority in the data classification under low target dimensions, which not only meets the requirements of reducing data dimensionality, but also maintains high classification performance.

From Figure 7, the convergence performance of selected target dimensions is as follows: From the horizontal comparison, we find that the convergence curves under these four dimensions tended to be stable after the 10th iteration. It can be observed that RMTSISOM-SPDDR also converged fast on Traffic. Similarly, we find, from the horizontal comparison, that the loss function still decreased under different dimensions, which is reasonable.

### 6.4. The Experiment on the ICT-TV Database

#### 6.4.1. The Description of ICT-TV

The ICT-TV Database [42] consists of two video datasets of TV series, namely, “Big Bang Theory” (BBT) and “Prison Break” (PB), which are shown in Figure 8 and Figure 9. ICT-TV consists of facial video shots taken in these two TV series, namely, 4667 shots in BBT and 9435 shots in PB. The pictures in the shots are stored with the size of 150 × 150 pixels. [30] were followed to preprocess these datasets. We adjusted all detected facial images to 48 × 60 pixels, then adopted histogram equalization, so as to reduce the effect caused by light. All facial frames were flattened into vector representations, on which PCA was performed. Hence, a set of 100-dimensional vectors were obtained which were applied to generate symmetric positive definite kernel matrices. For classification, we utilized the facial videos of 5 main characters in BBT and 11 main characters in PB, respectively.

#### 6.4.2. The Experimental Results on ICT-TV

Because the size of the datasets (BBT and PB) belonging to the ICT-TV database was excessively large, we segmented them to conduct experiments. We divided BBT (or PB) evenly into 10 sub-datasets. For each sub-dataset, training set and test set were divided from these samples with the ratio of 1:1. Finally, 10 experiments were conducted on these sub-datasets and took average results as final classification accuracies.

As can be seen from Figure 10, all algorithms showed high classification accuracies on BBT, while the result of RMTSISOM-SPDDR is higher than that of comparison algorithms in most dimensions. With the decrease in the target dimension, the performance of algorithms decreases. This may indicate that the lower the dimension is, the more information of the original data is lost, leading to a decline in classification.

The results of classification experiments on PB are shown in Figure 11. We can see that RMTSISOM-SPDDR is the most outstanding. This is mainly reflected in the fact that, under most target dimensions, our algorithm achieves the highest classification accuracy, while in the case of d=20, the accuracy of SSPDDR is the highest, followed by ours.

Convergence diagrams of RMTSISOM-SPDDR on BBT are shown in Figure 12. They indicate that our algorithm basically reflects preferable convergence performance. For each dimension, the curve was highly similar, generally leveling off after 15–20 iterations and there was almost no fluctuation.

Convergence diagrams of RMTSISOM-SPDDR on PB are shown in Figure 13. It is similar to the cases on BBT. The longitudinal comparison indicates that fifteen iterations were enough to bring the loss function down to an approximately minimum level and the fluctuation since then was very little.

### 6.5. The Experiment on the ETH-80 Dataset

#### 6.5.1. The Description of ETH-80

ETH-80 is a benchmark image set for recognition [43]. It covers eight categories (cars, cups, horses, dogs, cows, tomatoes, peas, apples). One category is comprised of ten objects, each of which consists of 41 pictures taken from different angles of an object, adjusted to 43 × 43 pixels. As shown in Figure 14, objects of the same category are different in terms of physical attributes, such as visual angle, color, texture distribution and shape. For this dataset, we made it into SPD data according to [24]. Specifically, let $M_{i}$ be the mean image of the $i^{th}$ image set, $M_{i}=\frac{1}{n_{i}}\sum_{j=1}^{n_{i}}C_{ij}$, where $n_{i}$ is the number of images in the $i^{th}$ image set, $C_{ij}$ is the $j^{th}$ image in the $i^{th}$ image set. Then, the SPD data corresponding to the $i^{th}$ image set can be calculated by $X_{i}^{*}=\frac{1}{n_{i}-1}\sum_{j=1}^{n_{i}}\left(C_{ij}-M_{i}\right){\left(C_{ij}-M_{i}\right)}^{T}$. Further, we added a correction term to $X_{i}^{*}$ so as to guarantee its positive definiteness, that is, $X_{i}=X_{i}^{*}+\frac{tr\left(X_{i}^{*}\right)}{\lambda }I$, $\lambda =10^{5}$. Each image set corresponds to an SPD matrix. In the experiment, we evenly divided image sets to training sets and test sets with the ratio of 1:1.

#### 6.5.2. The Experimental Results on ETH-80

It can be seen, form Figure 15, that, under each dimension selected, the classification accuracy of RMTSISOM-SPDDR is generally higher than that of the comparison algorithms and remains at a high level (above 80%). When RMTSISOM-SPDDR reduces the original dimension to different target dimensions, the fluctuation of classification accuracy is relatively small and the dimension can be reduced as much as possible without losing performance. Comparatively speaking, the results of DARG-graph and SSPDDR fluctuate greatly, which means there is a trade-off between classification accuracy and target dimension.

Figure 16 presents results of convergence experiments on ETH-80. The longitudinal comparison shows that the value of loss function gradually decreases with the increase in iteration times. Surprisingly, on ETH80, the second iteration leads to the largest drop and subsequent fluctuations are so small to be ignored. This may be because the initial projection matrix for this dataset was already near the local optimal solution. Moreover, the depression of the optimal solution is relatively deep, so that the results of subsequent iterations do not deviate from the optimal solution. From the horizontal comparison, we find that the loss function still tends to decrease under different target dimensions. Hence, our algorithm consistently shows excellent performance.

## 7. Discussions and Conclusions

(1)As a non-Euclidean data form, SPD data have in-depth applications in machine learning. Compared with vector representation, SPD data can more effectively extract higher-order statistical information. Generally, in order to avoid the problems of high computational complexity and sparse distribution in high-dimensional SPD manifolds, we hope to reduce the dimensionality of SPD data while maintaining useful and representative information. However, the nonlinear manifold structure results in difficulties in learning tasks where linear operations are needed.(2)SPD data equipped with a Riemannian metric generally constitute a Riemannian manifold. For Riemannian manifolds, we have known that all tangent spaces are complete finite-dimensional Hilbert spaces, namely, Euclidean spaces. Hence, this paper transfers learning tasks from original SPD manifolds to their tangent spaces.(3)The tangent space of an SPD manifold is a symmetric matrix space. From the perspective of data form and attributes, it can be proved to be the minimum linear extension of the original manifold. Inspired by this, we map SPD data into the tangent space at identity by the isometric transformation (*log*).(4)A framework is proposed for SPD data dimensionality reduction (DR). RMTSISOM-SPDDR realizes the procedure through the bilinear transformation between tangent spaces at identity matrices. These tangent spaces are Hilbert spaces and the required bilinear transformation can be determined to a specific criterion and then taken as the DR transformation for the original SPD manifold.(5)This paper specifies the bilinear transformation by the global isometric criterion. The so-called isometric criterion means that the geodesic distance on the original high-dimensional SPD manifold is equal to the corresponding Euclidean distance in the tangent space of the low-dimensional SPD manifold. This preserves the distance relationship between data points well.(6)In comparison to many existing state-of-the-art algorithms, such as PDDRSPD [34], PSSSD [36], DARG-Graph [37], SSPDDR [32] and LG-SPDDR [38], there are three differences between them and our proposed method (RMTSISOM-SPDDR). First, most of these algorithms are based on the bilinear transformation between two manifolds, while the proposed RMTSISOM-SPDDR is based on the bilinear transformation between the tangent spaces of the two manifolds. The tangent spaces are finite Hilbert spaces, i.e., Euclidean spaces, which can support more complex DR models. Second, all of these algorithms utilize local analysis, while the proposed RMTSISOM-SPDDR is based on global isometry, providing a different perspective. Finally, all of these algorithms are supervised, while the proposed RMTSISOM-SPDDR is unsupervised.

## Figures and Tables

**Figure 1 entropy-23-01117-f001:**
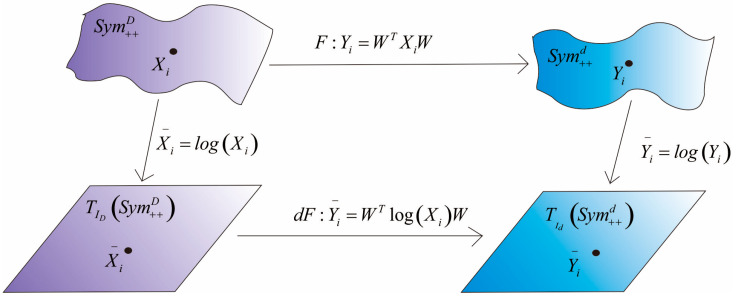
Schematic diagram of DR for SPD data. First, original SPD data are mapped to the tangent space at identity. Subsequently, the bilinear transformation between tangent spaces is determined in line with a specific criterion. Finally, the bilinear transformation learned from tangent spaces is taken to realize the DR of SPD data.

**Figure 2 entropy-23-01117-f002:**
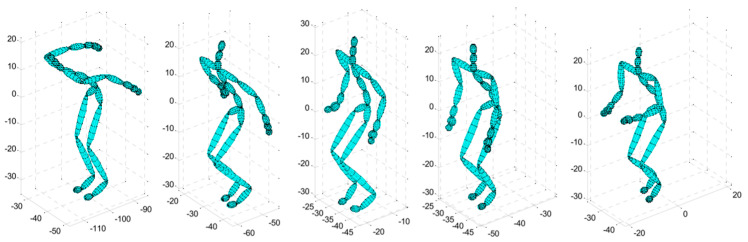
Samples of the Motion dataset.

**Figure 3 entropy-23-01117-f003:**
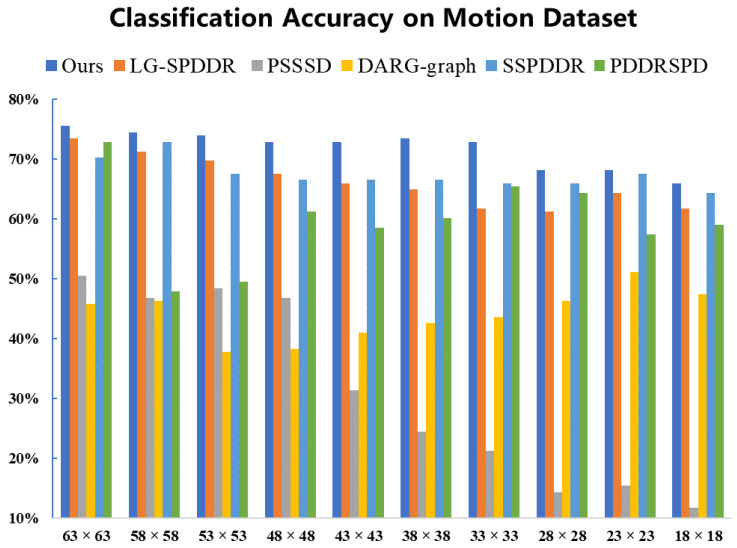
Classification accuracy on the Motion dataset. The horizontal axis represents the target dimension for DR and the vertical axis represents the classification accuracy.

**Figure 4 entropy-23-01117-f004:**
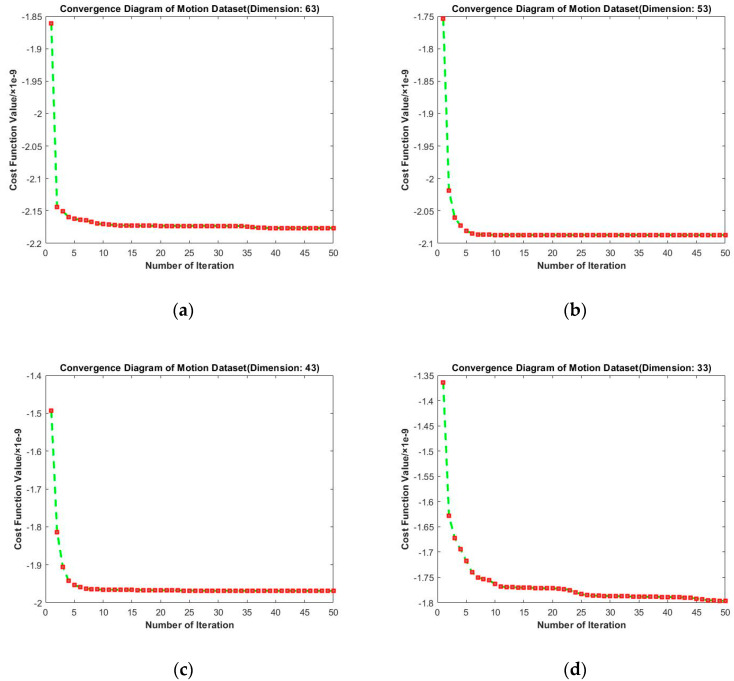
Convergence diagrams on the Motion dataset. The horizontal axis represents the number of iterations and the vertical axis represents the value of the loss function. (**a**–**d**) represent the results under different target dimensions (63, 53, 43, and 33) respectively.

**Figure 5 entropy-23-01117-f005:**
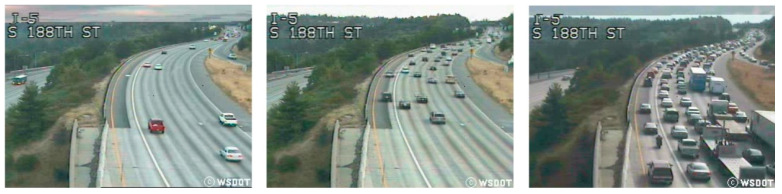
Samples of the Traffic dataset.

**Figure 6 entropy-23-01117-f006:**
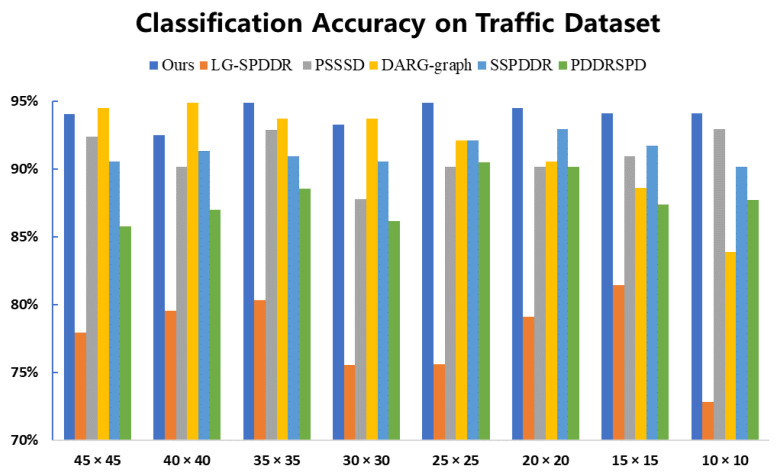
Classification accuracy on the Traffic dataset. The horizontal axis represents the target dimension for DR and the vertical axis represents the classification accuracy.

**Figure 7 entropy-23-01117-f007:**
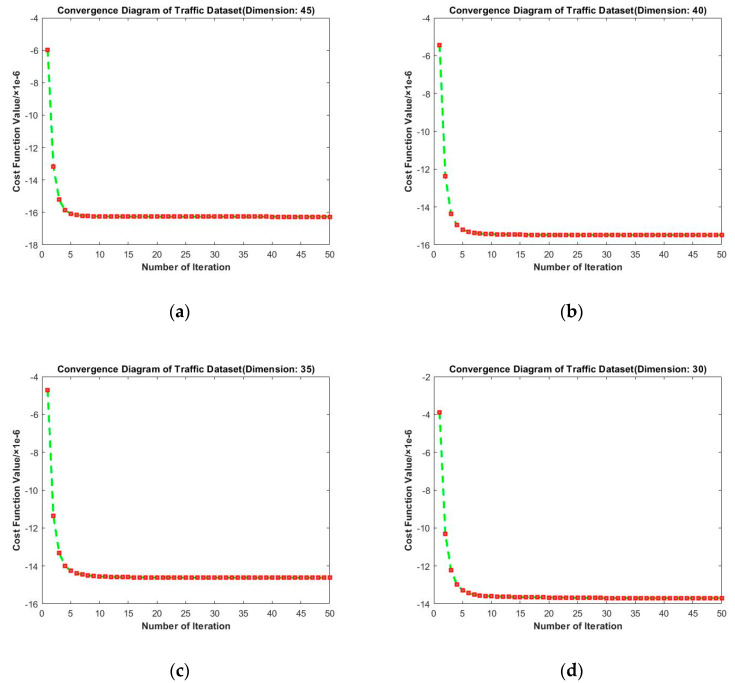
Convergence diagrams on the Traffic dataset. The horizontal axis represents the number of iterations and the vertical axis represents the value of the loss function. (**a**–**d**) represent the results under different target dimensions (45, 40, 35, and 30) respectively.

**Figure 8 entropy-23-01117-f008:**
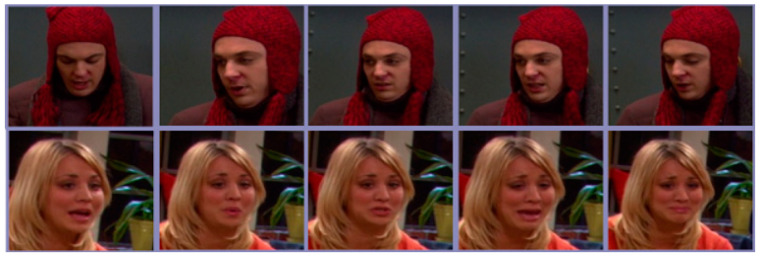
Samples of BBT Dataset.

**Figure 9 entropy-23-01117-f009:**
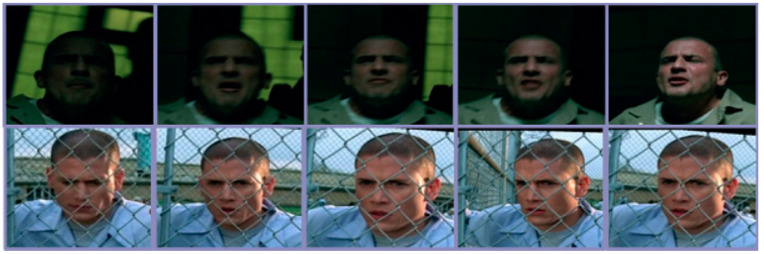
Samples of PB Dataset.

**Figure 10 entropy-23-01117-f010:**
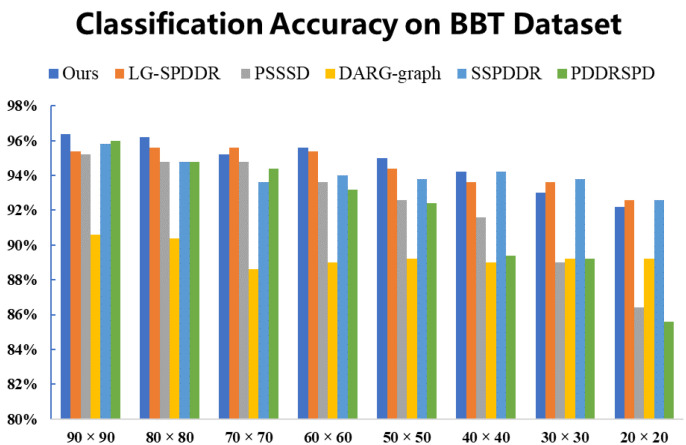
Classification accuracy on the BBT dataset. The horizontal axis represents the target dimension for DR and the vertical axis represents the classification accuracy.

**Figure 11 entropy-23-01117-f011:**
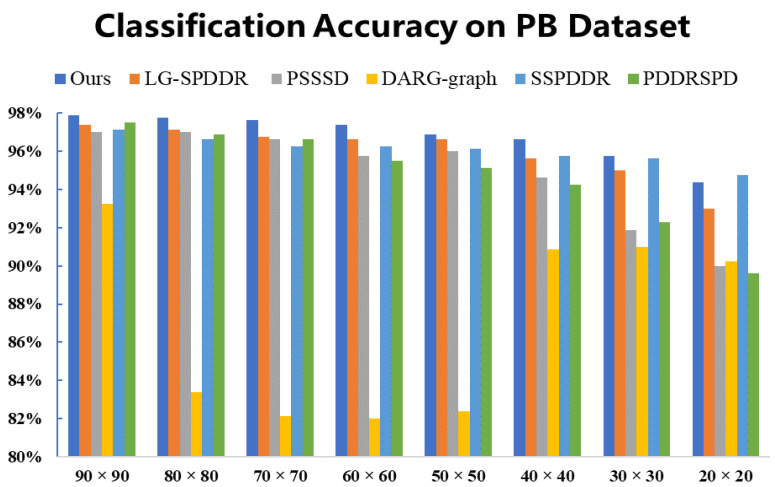
Classification accuracy on the PB dataset. The horizontal axis represents the target dimension for DR and the vertical axis represents the classification accuracy.

**Figure 12 entropy-23-01117-f012:**
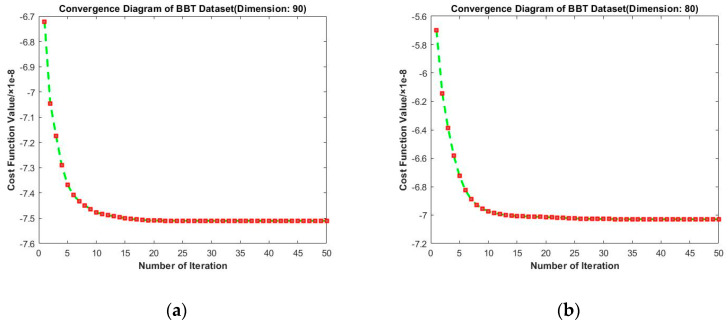
Convergence diagrams on the BBT dataset. The horizontal axis represents the number of iterations and the vertical axis represents the value of the loss function. (**a**–**d**) represent the results under different target dimensions (90, 80, 70, and 60) respectively.

**Figure 13 entropy-23-01117-f013:**
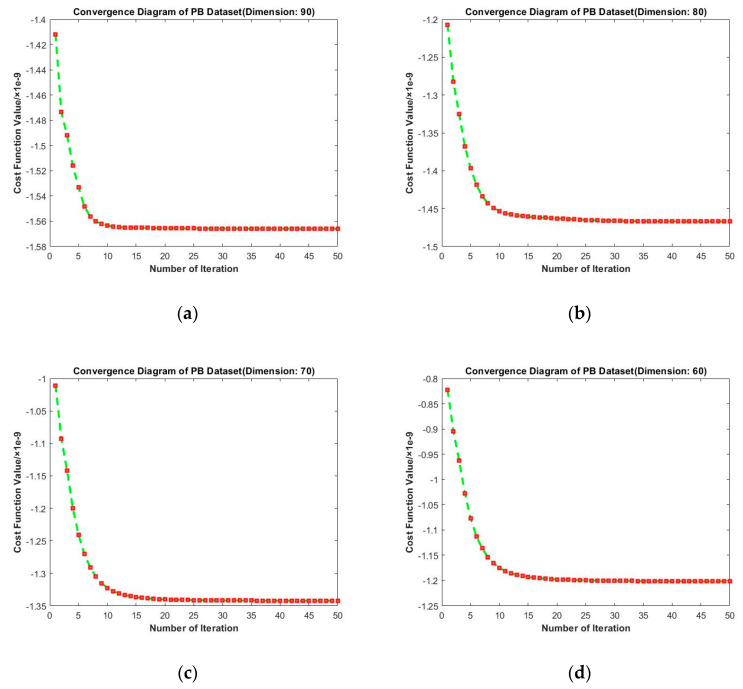
Convergence diagrams on the PB dataset. The horizontal axis represents the number of iterations and the vertical axis represents the value of the loss function. (**a–d**) represent the results under different target dimensions (90, 80, 70, and 60) respectively.

**Figure 14 entropy-23-01117-f014:**
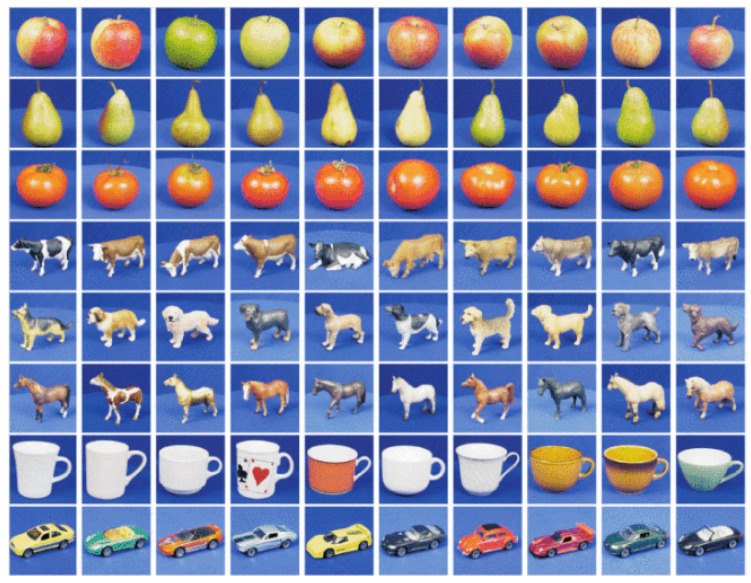
Samples of the ETH-80 dataset.

**Figure 15 entropy-23-01117-f015:**
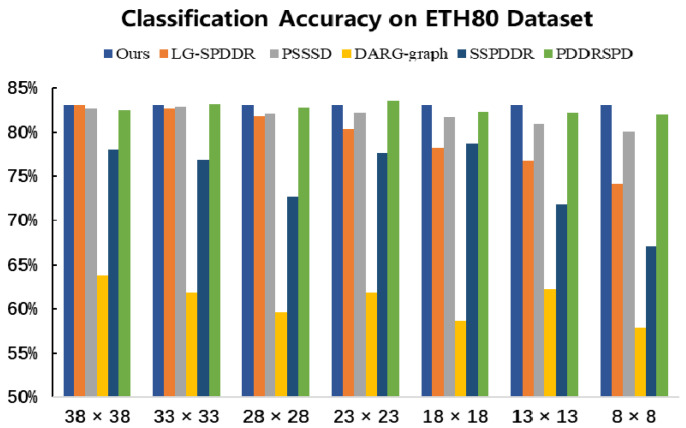
Classification accuracy on the ETH80 dataset. The horizontal axis represents the target dimension for DR and the vertical axis represents the classification accuracy.

**Figure 16 entropy-23-01117-f016:**
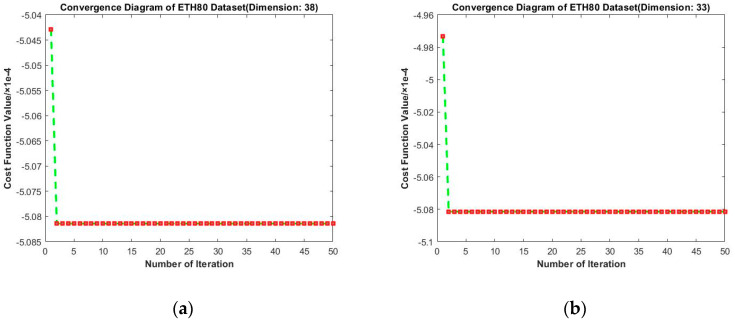
Convergence diagrams on the ETH80 dataset. The horizontal axis represents the number of iterations and the vertical axis represents the value of the loss function. (**a–d**) represent the results under different target dimensions (38, 33, 28, and 23) respectively.

**Table 1 entropy-23-01117-t001:** Notations and corresponding descriptions in this paper.

Notations	Descriptions
$T_{p}\left(M\right)$	The tangent space of the differential manifold M at p
$G_{p}\left(M\right)$	The differentiable function germ of the differential manifold M at p
$Sym_{++}^{D}$	The entire D×D SPD matrices
$Sym^{D}$	The entire D×D symmetric matrices
X	Capital letter denotes an SPD matrix
Φ	Greek capital letter denotes a tangent vector of an SPD manifold

**Table 2 entropy-23-01117-t002:** Parameter settings of five comparison algorithms.

Methods	Parameter Settings
SSPDDR	None
PDDRSPD	$\lambda_{w}=3$, $\lambda_{b}=3$, $\lambda_{s}=3$
DARG-Graph	σ=1
LG-SPDDR	$k_{w}=5$,$k_{b}=20$
PSSSD	$\zeta_{S}=1$, $\zeta_{D}=10$

## Data Availability

The data presented in this study are available on request from the corresponding author.

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
