# Peer review of "Dimensionality Reduction of SPD Data Based on Riemannian Manifold Tangent Spaces and Isometry"

_entropy, 2021, doi:10.3390/e23091117_

Round 1
Reviewer 1 Report
The paper is very interesting, all results seem are new and deserve publication.
Therefore, it could be accepted in the present form.
Reviewer 2 Report
The authors propose a new dimensionality reduction method based on tangent spaces of a (non-linear) Riemannian manifold. The introduction is relatively comprehensive but the section "7. Discussion (s) and Conclusion (s)" is excessively short. In particular, I think that it is necessary to provide a more detailed comparison with the existing algorithms in the conclusions.
On the other hand, the definitions in sections 2.2.1 are completely standard. Move it to an appendix and provide any of the standard references in the field. Finally, the result discussed in Appendix A is trivial (but English needs to be corrected). An appendix is ​​not necessary for such straightforward result.
Round 2
Reviewer 2 Report
The authors have complied with the requests of my report.
This manuscript is a resubmission of an earlier submission. The following is a list of the peer review reports and author responses from that submission.